# Photoacoustic imaging to localize indeterminate pulmonary nodules: A preclinical study

Chang Young Lee[ID][1,2][☯]*, Kosuke Fujino[1], Yamato Motooka[1], Alexander Gregor[1], Nicholas Bernards[1], Hideki Ujiie[1], Tomonari Kinoshita[1], Kyung Young Chung[2], Seung Hee Han[3][☯], Kazuhiro Yasufuku[1][☯]

1 Division of Thoracic Surgery, Toronto General Hospital, University Health Network, Toronto, Ontario, Canada, 2 Department of Thoracic and Cardiovascular Surgery, Yonsei University College of Medicine, Seoul, Korea, 3 Department of Medical Biophysics, University of Toronto, Toronto, Ontario, Canada

☯ These authors contributed equally to this work.
* cyleecs@yuhs.ac

## Abstract

**Data Availability Statement:** All relevant data are within the manuscript and its Supporting Information files.

### Purpose

Diagnosis and resection of indeterminate pulmonary nodules (IPNs) is a growing challenge with increased utilization of chest computed tomography. Photoacoustic (PA) -guided surgical resection with local injection of indocyanine green (ICG) may have utility for IPNs that are suspicious for lung cancer. This preclinical study explores the potential of PA imaging (PAI) to detect ICG-labeled tumors.

### Materials and methods

ICG uptake by H460 lung cancer cells was evaluated in vitro. A phantom study was performed to analyze PA signal intensity according to ICG concentration and tissue thickness/depth using chicken breast. PA signals were measured up to 48 hours after injection of ICG (mixed with 5% agar) into healthy subcutaneous tissue, subcutaneous H460 tumors and right healthy lung in nude mice.

### Results

Intracellular ICG fluorescence was detected in H460 cells co-incubated with ICG in vitro. The concentration dependence of the PA signal was logarithmic, and PA signal decline was exponential with increasing tissue depth. The PA signal of 2 mg/mL ICG was still detectable at a depth of 22 mm in chicken breast. The PA signal from ICG mixed with agar was detectable 48 hours post injection into subcutaneous tissue and subcutaneous H460 tumors in nude mice. Similar features of PA signals from ICG-agar in mice lung were obtained.

**Funding:** This research (Lee CY) was supported by Basic Science Research Program through the National Research Foundation of Korea (NRF) funded by the Ministry of Education (Grant number:2017R1D1A1B0302949714) and a faculty research grant of Yonsei University College of Medicine (Grant number:6-2016-0069). The funders had no role in study design, data collection and analysis, decision to publish or preparation of the manuscript.

**Competing interests:** The authors have declared that no competing interests exist.

## Conclusion

The results from this preclinical study suggests that PAI of injected ICG-agar may be beneficial for identifying deeply located tumors. These features may be valuable for IPNs.

## Introduction

With continuing improvements of imaging modalities and the increased application of low-dose computed tomography (CT) for lung cancer screening, more pathology with uncertain significance is detected in the lung parenchyma [1]. Indeterminate pulmonary nodules (IPNs) raise suspicion for lung cancer and should sometimes be surgically resected for diagnosis and treatment as CT-guided fine needle aspiration, or fluoroscopy-guided transbronchial biopsy can occasionally be inconclusive [2,3] with increased complication rates [4]. Although video-assisted thoracic surgery (VATS) is known to be ideal for IPN resection [5], identifying IPNs during VATS is somewhat challenging due to limited tactile perception, underscoring the need for preoperative localization [6]. New techniques for preoperative assessments have been introduced to improve success rates of planned resection and prevent unnecessary thoracotomy [7,8]. Fluorescence-guided lung resection has improved with the injection of agents such as indocyanine green (ICG) under preoperative guidance with navigation bronchoscopy [9,10], but it is hampered by limited penetration depth [11,12] and dispersion of injected contrast agents [13,14].

Photoacoustic imaging (PAI) is based on the photoacoustic (PA) effect, in which a laser is absorbed by endogenous or exogenous absorbers and is subsequently converted into heat, leading to thermoelastic expansion and thus generates acoustic waves which are detectable by a conventional ultrasound (US) transducer [15]. This enables deep tissue imaging while maintaining optical absorption contrast [16]. PAI can penetrate as deeply as 7 cm in tissue [17], which is sufficient for many clinical applications such as breast imaging or sentinel lymph node detection [18].

Agar has been used to localize IPNs either on its own [19] or mixed with dye such as methylene blue [20]. Although liquid agar can dissolve various dye or contrast agents, it rapidly solidifies upon local tissue injection.

In this preclinical study, we investigated the possibility of PA-guided surgery with local injection of ICG mixed with agar as a fiducial marker for the detection of IPNs.

## Materials and methods

### *In vitro* study

The human NSCLC cell line NCI-H460 (large cell carcinoma) was purchased from the American Type Culture Collection (Rockville, MD, USA). Cells were cultured and maintained as previously described by our laboratory [21]. H460 cells were cultured in humidified incubators at 37°C and 5% $CO_2$. Park Memorial Institute 1640 medium (Life Technologies Inc, Carlsbad, CA) was used. The media were also supplemented with 10% heat-inactivated fetal bovine serum (FBS), 50 U/mL penicillin, and 50 mg/mL streptomycin (Pen Strep, Life Technologies Inc). The cultured cells in one dish (BD Biosciences, Franklin Lakes, NJ) were incubated with two drops of NucBlue®LiveReadyProbes (NucBlueTM, Thermo Fisher Scientific, Waltham, MA) and ICG (IC green®, Akorn, Lake Forest, IL) of $3.2 \times 10^{-9}$ M/mL of medium for 30 minutes at 37°C under humidified air with 5% $CO_2$. The cells in the other dish were incubated

with NucBlue alone. The incubated cells were washed three times with phosphate-buffered saline to remove both free ICG and NucBlue. The cells were then retrieved by 10-minute exposure to 0.25% trypsin-ethylenediaminetetraacetic acid (Life Technologies Inc) to make cell suspensions. 100μl of the cell suspensions into a cuvette and centrifuged at 800 rpm for 3 minutes using Cytospin 3 (ShadonTM) to make cytologic slides. Those fluorescently stained cytologic slides were examined under Yokogawa spinning disk confocal microscopy (Carl Zeiss, Oberkochen, Germany). A DAPI specific laser (510–540 nm) was used to visualize nuclei stained with NucBlue, and an infrared (IR) wavelength laser (710–785 nm) was used to assess the presence of possible endogenous absorbers in H460 cells responding to near IR (NIR) light.

## Optical properties of ICG

The absorption spectra of ICG were measured using Cary 300 Bio UV-Visible Spectrophotometer (Agilent Technologies, Santa Clara, CA). After ICG powder was dissolved in distilled water, samples were diluted with distilled water to prepare six different ICG concentrations from 0.20 to 1.20 ug/mL. The values of attenuation coefficient for each concentration of ICG were obtained at 1-nm intervals from 600 to 900 nm. Bovine serum albumin (BSA, Sigma-Aldrich, St. Louis, MO) and purified agar powder (Sigma-Aldrich) were used as solutes to observe the changes in the ICG absorption spectrum according to solute type.

## Phantom study

Multiple blood vessel mimicking polymeric tubes were installed into Vevo®PHANTOM (Visualsonics, Fujifilm, Tokyo, Japan). ICG solutions mixed with distilled water and 20 mg/mL BSA were diluted to obtain eight different concentrations ranging from 0.125 to 5.0 mg/mL. These solutions were injected into the blood vessel mimicking tubes that were subjected to PAI in combination with conventional US (LZ-250, 20-MHz linear array transducer). Chicken breast muscle (6-mm-thick) which was purchased from grocery market was then placed over the tubes, the PA signal from ICG in the tubes was acquired in the range from 680 to 970 nm using a Vevo®LAZR system (Visualsonics, Fujifilm). To determine the maximum depth of signal penetration, 2.0 mg/mL of ICG in 50-uL tubes was placed at different depths. ranging from 0 to 22 mm deep)

## *In vivo* study

All animal experiments were approved by and conducted in accordance with the Toronto General Animal Care Committee under Animal Use Protocol 5908.01. All surgery or procedure was performed under isoflurane anesthesia, and all efforts were made to minimize suffering. We inoculated a mixture of 100 ul of Matrigel$^{TM}$ and H460 cells ($1x10^6$ cells/mouse) in the subcutaneous right flank tissue of nude female athymic mice (n = 3, Ncr-nu age 6–8 weeks; Taconic Farms Inc, Hudson, NY). Mice were monitored once a day by veterinary technicians using body conditioning scoring. Tumor formation was assessed daily, and tumor size was measured using an electronic caliper until the maximum tumor diameter reached 15 mm. Purified powdered agar was mixed with distilled water at a concentration of 5%. The agar powder was dissolved at 90˚C using a microwave. Liquid agar was kept warm and transferred into a tube that was placed in hot water (>50˚C) so that the agar would remain in its liquid form until injection. A 200 uL mixture of 2.0 mg/mL ICG and liquid agar was injected into healthy subcutaneous tissue in the left flank in 3 mice. Another 200 uL mixture of 2.0 mg/mL ICG and liquid agar was injected deep into subcutaneous tumor in the right flank in 3 mice. PA signals were measured up to 48 h after injecting the ICG-agar mixture. In order to identify the characteristics of ICG-agar mixture in lung tissue, 100 ul mixture of 2.0mg/mL ICG-agar was injected

into right lung of nude mice (n = 2) under ultrasound guidance. 8 mice in total were used in this study and cage enrichment was provided by group housing, nestlets and PVC tubing for all mice. After all experiments, mice were euthanized with $CO_2$.

### Histologic assessment

After mice were euthanized, the tumors and surrounding subcutaneous tissue were extracted. The tissue samples were fixed with formalin and stained with hematoxylin and eosin (H&E) for observation under bright-field microscopy to assess for histologic changes following ICG-agar mixture injection.

### Statistics

All graphs were plotted using R statistics (version 3.4.1).

## Results

### *In vitro* study

In both groups (NucBlue vs NucBlue + ICG), nuclei stained with NucBlue were detected under the DAPI wavelength of confocal microscopy. No signal was detected in H460 cells without ICG incubation under the IR wavelength while ICG uptake in the cell membrane or cytoplasm were detected in ICG incubated H460 cells under the IR wavelength (Fig 1).

### Absorption spectrum of ICG

ICG with a concentration of 0.2 ug/mL to 1.0 ug/mL dissolved in distilled water showed peak absorption at 780 nm (S1 Fig). The concentration dependence of ICG absorption was linear, expressed by the equation y = 0.402x + 0.045 (R2 = 0.985), where y represents the ICG absorption measured as attenuation coefficient and x represents the concentration of ICG in ug/mL (S2 Fig). The spectrum of ICG varied depending on the kinds of solute. ICG dissolved in distilled water showed peak absorption at 780 nm, while ICG mixed with BSA or agar showed peak absorption at 800 or 820 nm, respectively. Interestingly, unlike other substances, ICG mixed with agar showed a second peak at 880 nm (S3 Fig).

### Phantom study

The PA signal from each tube is displayed in red with the corresponding US signal in Fig 2A. The spectra of PA signal of ICG mixed with BSA, acquired in the range of 680 to 970 nm, are illustrated in Fig 2B. Although there was a slight ICG concentration dependent difference in the spectra of PA signal, two different peak signal intensities were observed at around 700 and 800 nm. The average PA intensity measured at 800 nm was calculated in selected regions of interest and then plotted against the ICG concentration in mg/mL (Fig 2C). The PA signal intensity of ICG was logarithmically dependent on the concentration and plateaued at 2.0 mg/mL. The intensities of the PA signal for 2.0 mg/mL ICG in chicken breast muscle are shown in Fig 2D. We observed an exponential decline in the PA signal intensity with increasing muscle depth (d, mm). The intensity of the PA signal measured at a thickness of 22 mm was 40 times weaker than the signal measured at 3 mm, but it could still be visually confirmed on the PA image and corresponding US image (Fig 3). The PA signal from ICG mixed with agar was also measured. Agar itself has no meaningful PA signal, and ICG mixed with agar showed a second peak of PA intensity at ~860 nm (S4 Fig).

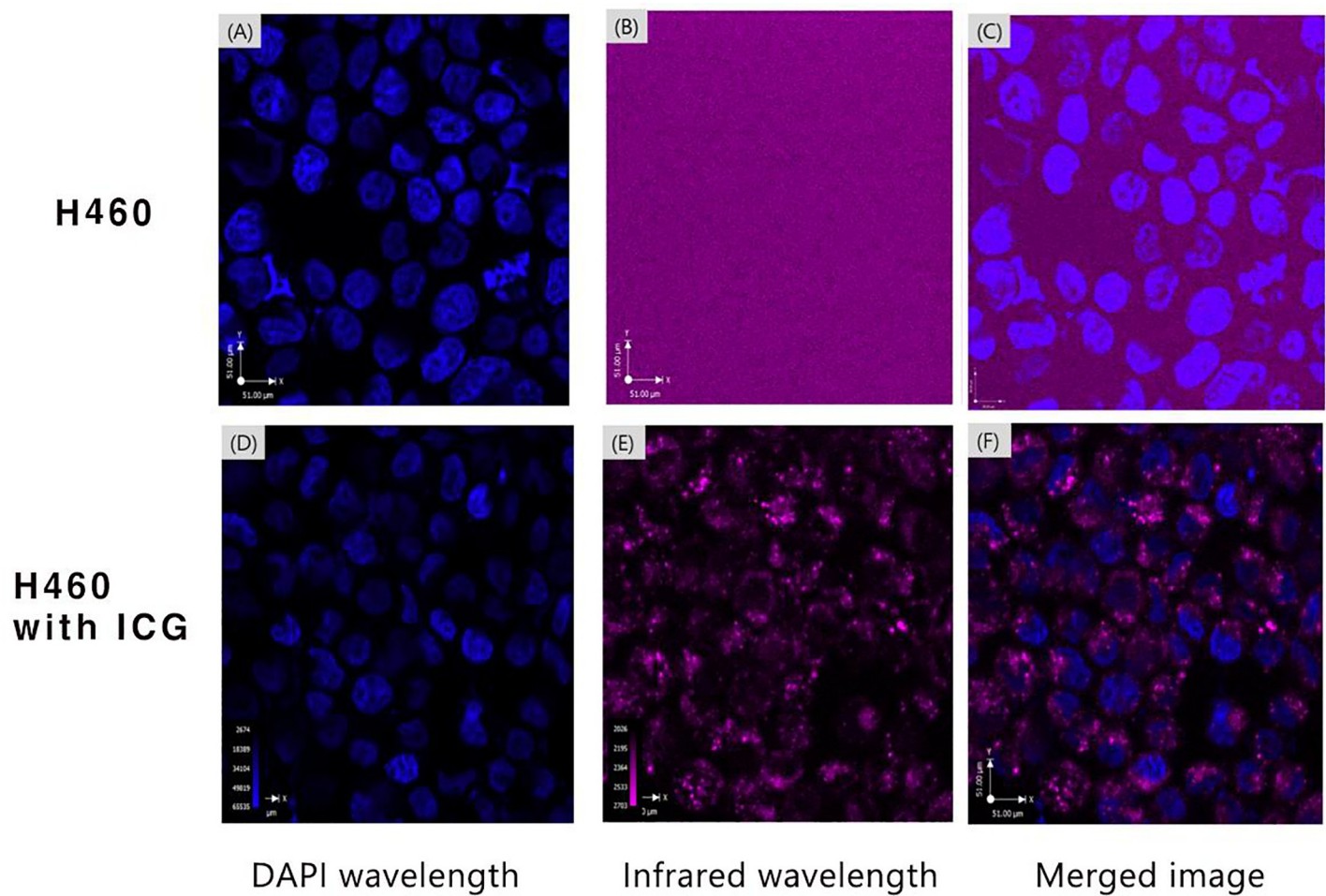

**Fig 1.** The images of H460 cells without ICG (A-C) and with ICG incubation (D-F) using confocal microscopy.

### *In vivo* study

ICG-agar injected into healthy subcutaneous tissues in mice maintained its shape on US even 24h and 48 h post injection (Fig 4A–4C). Conversely, the PA signal pattern from injected ICG-agar changed over time. The PA spectrogram taken immediately following injection showed strong signal at 800 and 900 nm (Fig 4A), similar to the pattern we observed in the phantoms using the vessel mimicking tubes. At 24 h after injection, the PA signal at 800 nm gradually weakened (Fig 4B), leaving only a 900 nm peak at 48 h (Fig 4C). In addition, the strong signal at 900 nm on the PA image from ICG-agar injected deep into H460 tumors in mice was visible at 48 h, even if no clear lesion was observed with US (Fig 4D). PA spectrogram obtained 48 h after injection into mice lung showed similar pattern with that of subcutaneous model which has a 900nm peak (Fig 5A). Even if PA signals from blood (dark orange), which contains oxy or de-oxyhemoglobin, was detected in the layer of chest wall muscle, PA signal from ICG-agar (green) in right lung of mice was clearly seen on transverse overlay image (Fig 5B) and 3D rendered image (Fig 5C).

### Histologic assessment

The boundary between the injected ICG-agar and H460 tumor was clearly visible in resected specimens (Fig 6A), and these delineations correlated well with PA images (Fig 6B) even

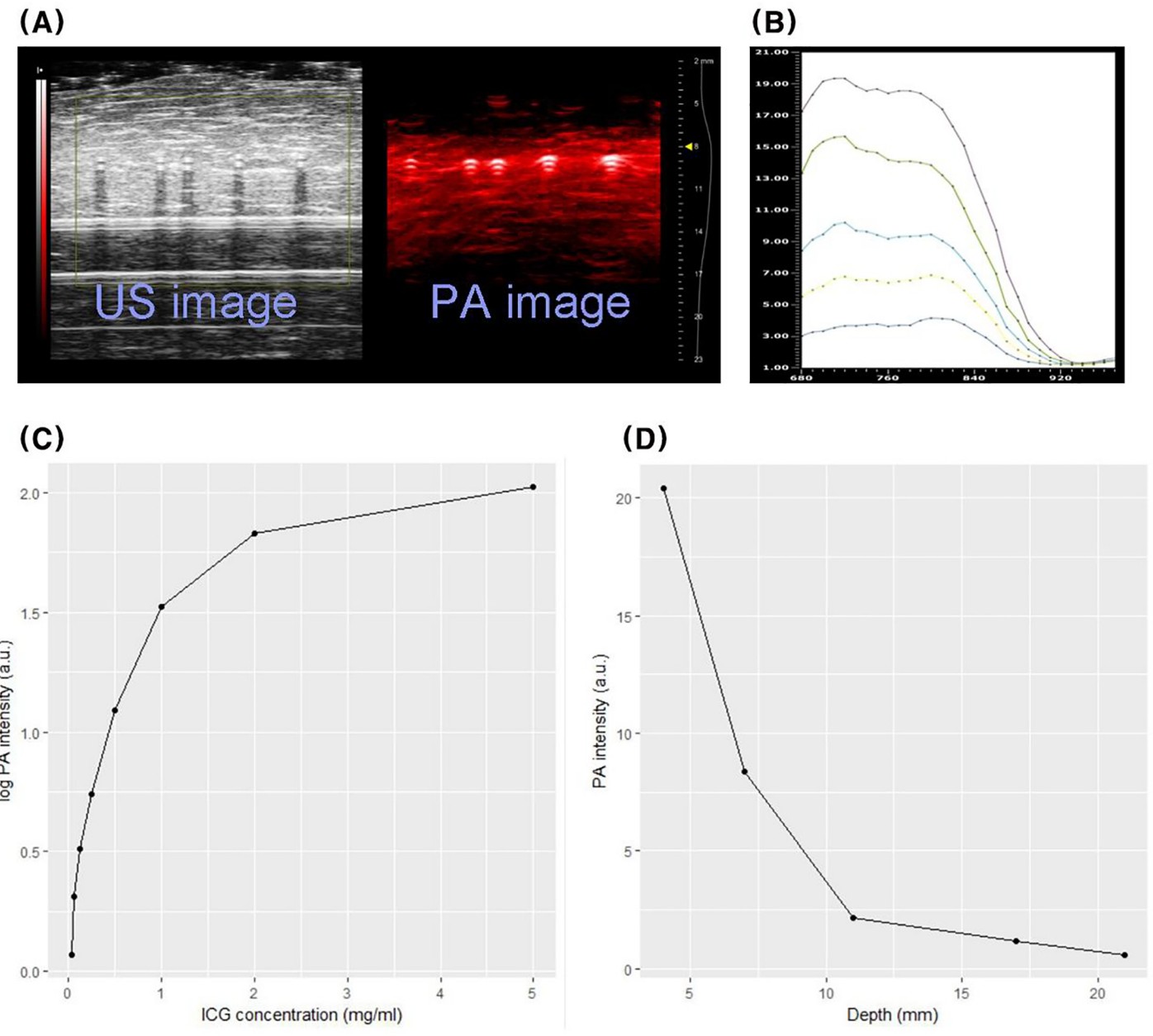

**Fig 2.** A. US and PA images from ICG at various concentrations. B. PA spectrum at various concentrations. C. The graph of PA signal intensity depending on ICG concentration. D. Depth dependence PA signal from ICG in chicken breast muscle phantom.

though the boundary between tumor and ICG-agar was not clear on US (Fig 6C). There was none or little ICG-agar remaining in H&E-stained tissue after fixation with formalin, but the boundary of the tumor was clearly observed, and normal morphology was maintained in the surrounding healthy muscle tissue (Fig 6D and 6E).

## Discussion

With the growing application of low-dose chest CT for lung cancer screening and regular check-ups [1,22], the detection rate of small IPNs has gradually increased. Localization during

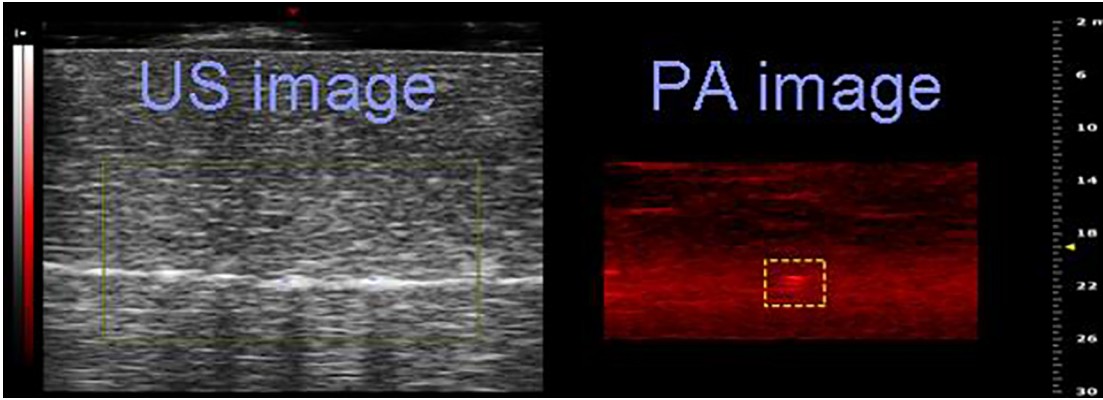

**Fig 3. US and PA image measured at 22mm depth from 2.0mg/ml ICG.**

VATS resection is very difficult for non-visible nodules or non-palpable, which prolongs operation time. Preoperative localization techniques have been introduced to improve the success rates of planned resection and prevent unnecessary thoracotomy [7,8].

Among various localization methods, fluorescence-guided lung resection has improved with the injection of fluorescent agents, such as ICG under guidance with navigation bronchoscopy [9,10]. This method has advantages in reducing potential complications such as symptomatic pneumothorax or hemothorax and avoiding radiation exposure during CT-guided transthoracic localization [23,24]. However, limitations include limited penetration depth and dispersion of injected contrast agents.

Due to the scattering and absorbing nature of biological tissue, optical fluence decays rapidly with increased depth [16]. As a result, the amount of light reaching the region of interest located further away from the laser source may be insufficient to detect deeply located tumors. PAI is a nascent imaging technology based on the PA effect that was discovered by Alexander Graham Bell in 1880 [17]. This effect is due to the formation of sound waves following light absorption in an endogenous (e.g., melanin or hemoglobin) or exogenous (e.g., ICG or nanoparticles) absorber. The generated US wave can be detected with a conventional US transducer that enables deep tissue imaging while maintaining optical absorption contrast. This approach enables strong optical contrast in optically scattered biological tissue at depths <1–5 cm depending on the laser wavelength [25], optical properties of absorbers [18] and characteristics of the acoustic transducers [26].

Given IPNs deep location at 2–3 cm from visceral pleura, we employed an NIR wavelength laser for improved depth penetration. Besides melanoma, most cancer cells have few or no endogenous absorbers that can absorb the NIR wavelength [27]. This was confirmed by our confocal experiments using H460 human lung cancer cells. However, ICG incubated with H460 cells was clearly detectable with the NIR wavelength and confirmed the possibility of using ICG as an exogenous PA contrast agent [28].

Because ICG itself has a relatively low quantum yield (0.027, referring to fluorescence emission) [28], it has been adopted for cancer research studies using PAI to identify sentinel lymph nodes. However, the hydrophobicity of ICG can result in substantial concentration and environment-dependent changes in optical properties, as well as photo-instability. ICG-encapsulated nanoparticles have been explored as an option to overcome these limitations. However, one report stated that that PA signal from ICG liposome was clearly resolved to a depth of 10 mm using chicken breast muscle [29], which is not appropriate for IPN localization. Our phantom study demonstrated that the PA signal increased logarithmically without a

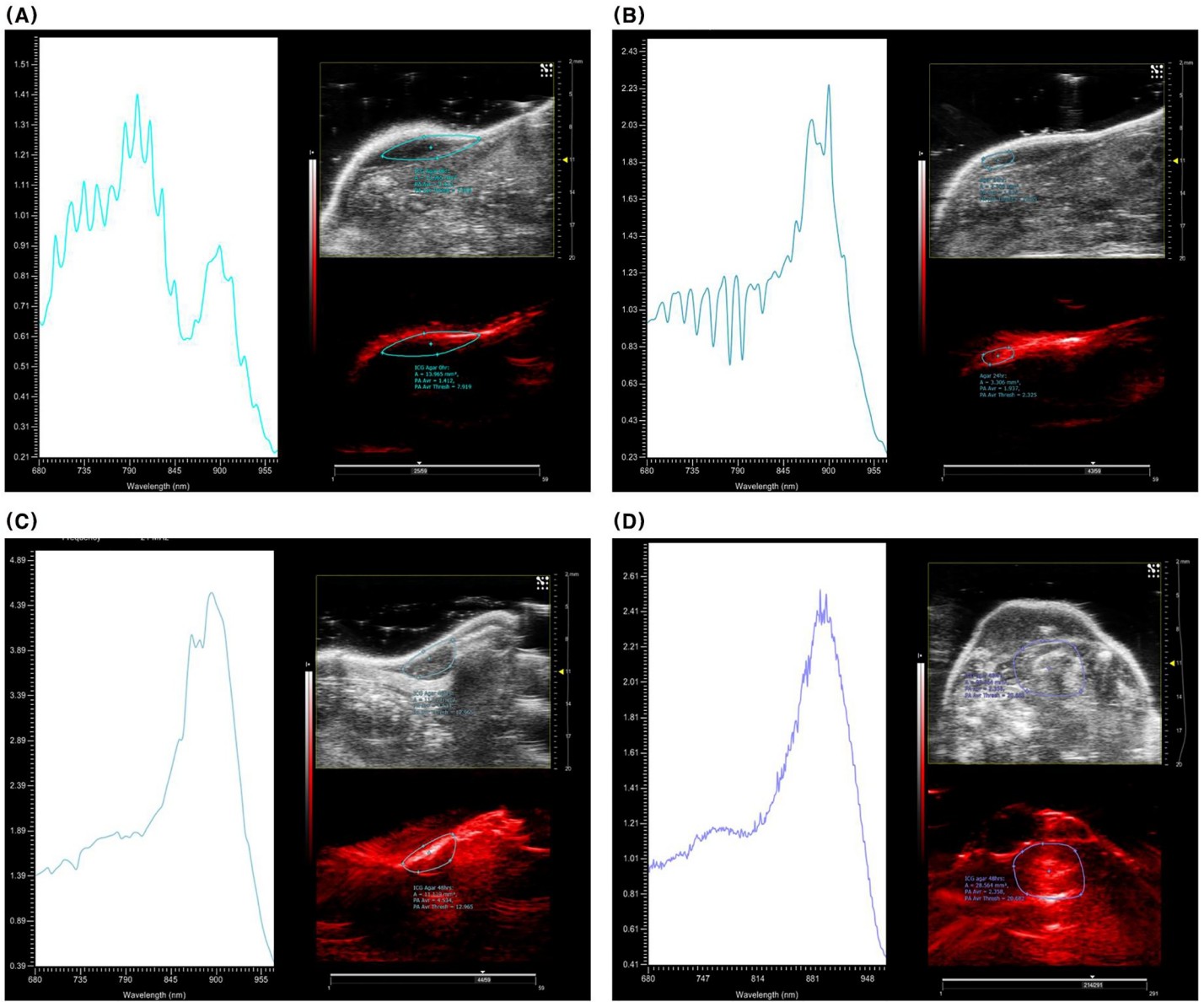

**Fig 4.** US and PA images and corresponding PA spectrogram from injected ICG-agar into healthy subcutaneous tissue at 0hr (A), 24hr (B), and 48hr (C) Fig 4D US and PA image and spectrogram obtained from ICG agar injected deep to subcutaneous tissue H460 tumor (48hr).

quenching effect, which decreases the fluorescence intensity at higher ICG concentrations [9], up to 5 mg/mL. Even if an exponential decline of the PA signal intensity at a concentration of 2.0 mg/mL ICG was observed at greater depths of chicken breast muscle, the signal intensity at a depth of 22 mm could still be visible via PAI. This suggests that PAI using a high concentration of ICG would be appropriate for localizing deep IPNs.

Agar is known to retain its shape in the lung, with or without contrast agents [19,20]. We found that agar mixed with ICG maintained its shape on US images over 48 h after injection, but the PA spectrogram of injected ICG-agar was different than that of ICG mixed with BSA. This phenomenon seems to be related to the temperature dependent optical property of ICG. Mauerer et al. [30] reported temperature dependent J-aggregation at high concentrations of

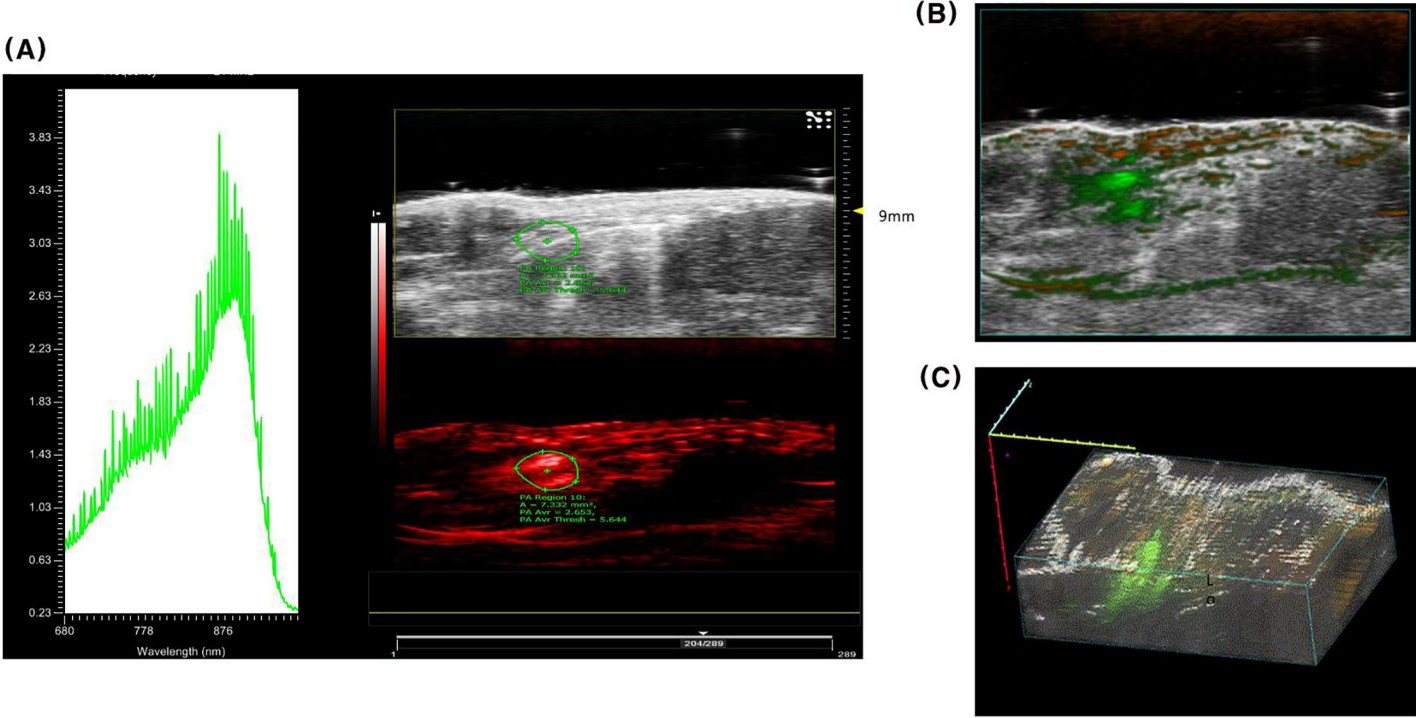

**Fig 5.** A. PA spectrogram obtained from ICG-agar 48hr after injection in mice lung. B. Transverse overlay image of mice lung with injected ICG-agar. C. 3D rendered image of mice lung with injected ICG-agar.

ICG, resulting in the absorption peak at 893 nm. ICG-agar, which was mixed in agar solution kept at 50ºC or higher just before injection, might cause a change of absorption spectrum. Furthermore, an altered pattern in the ICG-agar PA spectrogram was observed over time not only subcutaneous tissue but lung tissue in mice. We propose that the PA signal from J-aggregated ICG bound with agar might only persist for 48 h after injection, although the effect of PA signal from free ICG, which would be drained into the lymphatic system over time, was strong immediately post ICG-agar injection.

One of the desired characteristics of liquid fiducial markers is to have none or a limited effect on the tumor and surrounding tissues. Agar does not change the histologic morphology of lung tumors and surrounding tissues, either by itself [19] or after being mixed with methylene blue [20]. Although ICG can accumulate by passive tumor cell-targeting or the inherent ability of ICG to bind cell membranes, it does not affect the overall morphology or pathologic diagnosis [10,12]. Resected specimens in this study also had clear tumor margins from injected ICG-agar, and this correlated with the PA images despite the boundary being unclear on conventional US. H&E staining after fixation with formalin, which dissolved ICG-agar, did not reveal morphological changes at the tumor margin or in the surrounding tissue.

There are some limitations of this study. Although this study has been comprehensively covered in vitro, phantom and in vivo experiments using subcutaneous mice and lung mice model, but e did not examine larger animals. Further studies are therefore necessary to confirm the possibility of PAI for IPN localization using rabbits or pigs. Unlike other studies that reported a maximum penetration depth of 7 cm [17], our maximum penetration depth was shallow at 22 mm, despite the use of high ICG concentration. This is likely due to the LZ 250 (20-MHz linear array transducer) used in this study that has an optimum target depth of 10–15 mm, which is well suited for preclinical applications [26]. Future investigations should use a

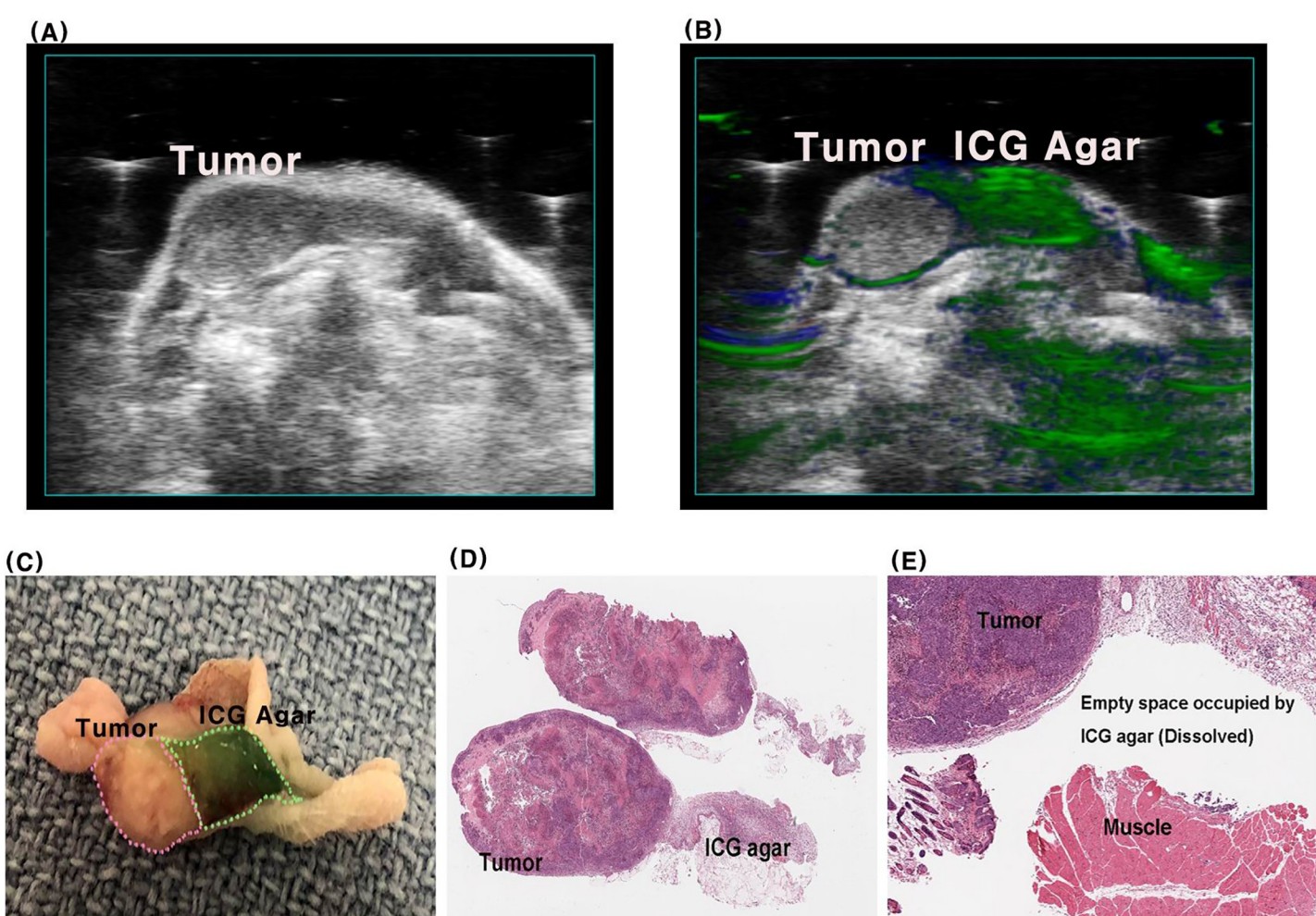

**Fig 6. Longitudinal cross section of subcutaneous tumor and ICG-agar after 48hr after injection.** A: US image, B: US and PA overlay image, C: Resected specimens, D, E:H&E stained slide.

low-frequency transducer to identify lesions that are ≥3 cm deep. In addition, the PA spectrum of ICG-agar changed over time, even if the spectral properties remained stable 48 h after injection. Further studies using different kinds of liquid fiducial markers, such as lipiodol or PalpMark™ for example, should be performed to minimize PA spectral changes over time.

## Conclusions

This preclinical study suggests that PAI of injected ICG-agar may be useful for localizing deeply located tumors. This could be valuable for IPNs. ICG-agar maintained its shape up to 48 h post tissue injection without disrupting the histology of the tumor or surrounding tissue. The PA properties of ICG changed over time but remained detectable on PAI. These results should be validated with ICG-agar injection into large animal models such as rabbit or pig.

## Supporting information

**S1 Fig. The absorption spectrum of ICG dissolved in distilled water according to concentration change.**
(TIF)

**S2 Fig. Calibration curve for ICG in distilled water, absorbance measured at 780nm.**
(TIF)

**S3 Fig.** ICG absorbance in distilled water (blue, dashed line), bovine serum albumin (orange, solid line) and agar (green, dotted line).
(TIF)

**S4 Fig. The PA spectrogram (left), US image (right upper) and PA image (right lower).**
Red line shows PA spectral change of ICG agar and green line shows that of agar itself in the right image.
(TIF)

**S1 Data set.**
(ZIP)

## Author Contributions

**Conceptualization:** Chang Young Lee, Tomonari Kinoshita, Kyung Young Chung, Seung Hee Han, Kazuhiro Yasufuku.

**Data curation:** Chang Young Lee, Yamato Motooka, Alexander Gregor, Nicholas Bernards, Hideki Ujiie, Seung Hee Han, Kazuhiro Yasufuku.

**Formal analysis:** Chang Young Lee, Yamato Motooka, Alexander Gregor, Nicholas Bernards, Seung Hee Han.

**Investigation:** Kosuke Fujino, Yamato Motooka, Alexander Gregor, Nicholas Bernards, Tomonari Kinoshita.

**Methodology:** Kosuke Fujino, Alexander Gregor, Nicholas Bernards, Hideki Ujiie, Tomonari Kinoshita.

**Software:** Hideki Ujiie.

**Supervision:** Kyung Young Chung, Seung Hee Han, Kazuhiro Yasufuku.

**Visualization:** Tomonari Kinoshita, Seung Hee Han.

**Writing – original draft:** Chang Young Lee, Kosuke Fujino, Kyung Young Chung, Seung Hee Han, Kazuhiro Yasufuku.

**Writing – review & editing:** Chang Young Lee.

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
