## [Decision Letter · Decision Letter 0]

2 Jan 2020

PONE-D-19-30111

Photoacoustic Imaging to Localize Indeterminate Pulmonary Nodules: A Preclinical Study

PLOS ONE

Dear Dr Lee,

Thank you for submitting your manuscript to PLOS ONE. After careful consideration, we feel that it has merit but does not fully meet PLOS ONE’s publication criteria as it currently stands. Therefore, we invite you to submit a revised version of the manuscript that addresses the points raised during the review process.

We would appreciate receiving your revised manuscript by Feb 16 2020 11:59PM. To enhance the reproducibility of your results, we recommend that if applicable you deposit your laboratory protocols in protocols.io, where a protocol can be assigned its own identifier (DOI) such that it can be cited independently in the future. For instructions see: http://journals.plos.org/plosone/s/submission-guidelines#loc-laboratory-protocols

We look forward to receiving your revised manuscript.

Kind regards,

Gabriele Multhoff, Prof. Dr.

Academic Editor

PLOS ONE

Journal Requirements:

1. 

2.  Please provide additional information about the NCI-H460 cells used in this work, including any quality control testing procedures (authentication, characterisation, and mycoplasma testing). For more information, please see http://journals.plos.org/plosone/s/submission-guidelines#loc-cell-lines.

3.  At this time, we ask that you please provide the source of the chciken breast used in this study.

4.  At this time, we request that you  please report additional details in your Methods section regarding animal care, as per our editorial guidelines:

(1) Please state the number of mice used in the study  

(2) Please provide details of animal welfare (e.g., shelter, food, water, environmental enrichment)

(3) Please describe any steps taken to minimize animal suffering and distress, such as by administering anaesthesia, during the inoculation of H460 cells

(4) Please state the specific number of H460 cells that were subcutaneously inoculated into the mice

(5) Please include the method of euthanasia  

(6) Please describe the post-operative care received by the animals, including the frequency of monitoring and the criteria used to assess animal health and well-being.

Thank you for your attention to these requests.

5.  To comply with PLOS ONE submission guidelines, in your Methods section, please provide additional information regarding your statistical analyses. For more information on PLOS ONE's expectations for statistical reporting, please see https://journals.plos.org/plosone/s/submission-guidelines.#loc-statistical-reporting.

Reviewers' comments:

Reviewer's Responses to Questions

**Comments to the Author**

1. Is the manuscript technically sound, and do the data support the conclusions?

Reviewer #1: Partly

Reviewer #2: No

2. Has the statistical analysis been performed appropriately and rigorously? 

Reviewer #1: No

Reviewer #2: No

3. Have the authors made all data underlying the findings in their manuscript fully available?

Reviewer #1: Yes

Reviewer #2: Yes

4. Is the manuscript presented in an intelligible fashion and written in standard English?

Reviewer #1: Yes

Reviewer #2: No

5. Review Comments to the Author

Reviewer #1: The manuscript described photoacoustic (PA) imaging modality for possible detection of pulmonary nodules in lung, using ICG. ICG has been used in a variety of PA studies as an agent of choice, as standard in comparison studies, and hence authors judiciously chose it as a marker for their studies. The agar is solidfiable medium, injectable at high temperature that has been used in previous studies in pulmonary nodule detection, also taken in this study. Indeed agar has been used as phantom template by a prominent PA device manufacturer as tissue-mimicking phantom.

Hence the authors selection of agar + ICG as injectable medium for indeterminate pulmonary nodule is comprehendible. The authors indicate neither ICG nor agar affects the overall morphology or pathology of tissue, which atleast after 48h, proved by histology.

The PA generation at various concentration (ICG: 2.0 mg/mL) and at depth (upto 22 mm) was carried out which showed a logarithmic increase and exponential decrease, respectively. The authors performed phantom study with agar+ICG mixtures (as function of concentration), performed subcutaneous tumor (by intra-tumoral injection) and healthy tissue injection.

In addition, the material injected into the lung tissue and obtained PA signal after 48h. Can the PA signal be visible even after 48h, which was not carried out.

The paper is clearly well-written, yet lacks to bring up the innovation aspect. This reviewer failed understand the how the results of this study can be translated to detect pulmonary nodules. Hence an artistic (schematic) figure could help to understand how this preclinical PA imaging with injected ICG-agar predicts the localizing deeply located pulmonary nodules/tumors.

The following points need to be adressed:

1) As the PA imaging still limited with depth (and the US imaging of lung is also difficult) how you think the study can be translated.

2) 'Monomeric' ICG vs J-aggregated ICG in agar: Preparation of J-aggregated ICG was already reported (Chemical Physics 1997, 220 , 385-392). Could you please perform the characterization the latter species in agar.

3) Then, as you suggest draining characterics of 'mono ICG' was faster than 'J-aggr ICG', could it be proved by incubating both current formulation and the J-aggr ICG+agar pellet (a solidified agar+ICG) by incubating at 37C for 48h to 76h.

4) Overall the images quality is very poor; need to be post-processed to improve the quality; as well provide the spectra and graphs with clearly readable numbers.

5) Is there any study on blank agar effect on surrounding tissue of injection, please cite the pertinent references

6) The phantom image, Fig 2, extremely overlapped (high noise) does it cause signal spill-over, and over estimation. Can you comment.

7) Where is statistical evaluation of in vivo analysis.

8) Fig 5, the depth scale is missing.

9) It is known that ICG not so photostable; comment whether other reported 'stable' PA chromophores could be suited for this study.

10) Is it possible to show, in an animal model, guided injection of agar+dye in pulmonary nodule.

Reviewer #2: The main points of criticism of this manuscript pertain to (1) the confusion about the purpose of the study and (2) the unconvincing model systems used and the poor presentation of the data.

Ad 1) Even after reading the manuscript several times, it did not become clear to me which future application that preclinical study is supposed to prepare.

Is the ICG-containing agar supposed to represent a fiducial marker that is meant to aid surgical removal of suspicious nodules or is the idea that tumor cells could be labeled with ICG to facilitate their removal? For the latter case, the reader can only speculate on how selective targeting is supposed to work.

Ad 2) Numerous photoacoustic imaging studies are using ICG-labeled cells in tissue phantoms and animal models; some of them come close to a setting that could eventually be relevant for clinical applications.

None of the data presented in the current work, however, comes close to a realistic scenario of detecting ICG-labeled cells or fiducial markers (see the confusion due to conflicting information in the text described in point (1)) within lung tissue, which is an intrinsically complicated tissue to image by photoacoustics due to the high density of air-filled structures.

While in NCI-H460 cells, ICG seems to be only detected by fluorescence, neither the data from blood vessel mimicking tubes nor the chicken breast is convincing, new, or convincing.

The in vivo work was then apparently just conducted with ICG-containing agar as opposed to cells injected subcutaneously or into the lungs of mice: the data quality is bad, and it is not clear what is supposed to be learned from the data other than confirming yet one more time that highly-concentrated ICG can be localized in tissue by photoacoustics.

In general, the presentation of the spectra and images look like distorted screen-shots from a poor GUI. The scientific images are missing defined colormaps and scale bars. Just as one example, how was the color-overlay in Figure 6B computed?

There are also many linguistic weaknesses in the text.

I would thus like to encourage the authors to put more effort into presenting a consistent story and an adequate presentation of the data prior to re-submission of this work.

6. PLOS authors have the option to publish the peer review history of their article (what does this mean?). If published, this will include your full peer review and any attached files.

Reviewer #1: No

Reviewer #2: No

---

## [Author Response · Author response to Decision Letter 0]

18 Feb 2020

Journal Requirements:

-> I have rechecked additional requirements as you requested. 

2. Please provide additional information about the NCI-H460 cells used in this work, including any quality control testing procedures (authentication, characterisation, and mycoplasma testing). For more information, please see http://journals.plos.org/plosone/s/submission-guidelines#loc-cell-lines.

-> As guideline recommended, a reference and repository from another company was addressed in manuscript. 

3. At this time, we ask that you please provide the source of the chciken breast used in this study.

-> Chicken breast muscle was purchased from a grocery market. 

4. At this time, we request that you please report additional details in your Methods section regarding animal care, as per our editorial guidelines:

(1) Please state the number of mice used in the study 

-> 8 mice in total were used in this study. H406 cells were inoculated in 3mice. ICG agar was injected into another healthy 3mice and 2 mice was used for lung injection of ICG agar. 

(2) Please provide details of animal welfare (e.g., shelter, food, water, environmental enrichment)

-> Cage enrichment was provided by group housing, nestlets and PVC tubing for all mice. 

(3) Please describe any steps taken to minimize animal suffering and distress, such as by administering anaesthesia, during the inoculation of H460 cells

-> All surgery and procedure was performed under isoflurane anesthesia and all efforts were made to minimize suffering. 

(4) Please state the specific number of H460 cells that were subcutaneously inoculated into the mice

-> H460 cells were inoculated in 3 mice. 

(5) Please include the method of euthanasia 

-> After all experiemtns, mice were euthanized with CO2. 

(6) Please describe the post-operative care received by the animals, including the frequency of monitoring and the criteria used to assess animal health and well-being.

-> Mice were monitored by veterinary technicians once a day by veterinary technicians using body conditioning scoring.

Thank you for your attention to these requests.

5. To comply with PLOS ONE submission guidelines, in your Methods section, please provide additional information regarding your statistical analyses. For more information on PLOS ONE's expectations for statistical reporting, please see https://journals.plos.org/plosone/s/submission-guidelines.#loc-statistical-reporting.

-> All graphs were plotted using R statistics (version 3.4.1). 

-> I have updated my ORCID iD in Editorial Manager. 

Reviewer #1: The manuscript described photoacoustic (PA) imaging modality for possible detection of pulmonary nodules in lung, using ICG. ICG has been used in a variety of PA studies as an agent of choice, as standard in comparison studies, and hence authors judiciously chose it as a marker for their studies. The agar is solidifiable medium, injectable at high temperature that has been used in previous studies in pulmonary nodule detection, also taken in this study. Indeed agar has been used as phantom template by a prominent PA device manufacturer as tissue-mimicking phantom.

Hence the authors selection of agar + ICG as injectable medium for indeterminate pulmonary nodule is comprehendible. The authors indicate neither ICG nor agar affects the overall morphology or pathology of tissue, which atleast after 48h, proved by histology.

The PA generation at various concentration (ICG: 2.0 mg/mL) and at depth (up to 22 mm) was carried out which showed a logarithmic increase and exponential decrease, respectively. The authors performed phantom study with agar+ICG mixtures (as function of concentration), performed subcutaneous tumor (by intra-tumoral injection) and healthy tissue injection.

In addition, the material injected into the lung tissue and obtained PA signal after 48h. Can the PA signal be visible even after 48h, which was not carried out.

The paper is clearly well-written, yet lacks to bring up the innovation aspect. This reviewer failed understand the how the results of this study can be translated to detect pulmonary nodules. Hence an artistic (schematic) figure could help to understand how this preclinical PA imaging with injected ICG-agar predicts the localizing deeply located pulmonary nodules/tumors.

Thanks for your comments and some questions.

 When thoracic surgeons would encounter the patients with indeterminate pulmonary nodules (IPNs) which are not likely to being detected or palpated during minimally invasive surgery, various kinds of localization methods have been using. Recently transbronchial or transthoracic ICG injection around IPNs under CT or navigation bronchoscope guidance prior to operation has widely been accepted to see the fluorescence signal with near infrared camera during operation. However, there are two limitations in this procedure. First is penetration depth limitation of fluorescence signal that would hamper the localization of deeply located IPNs and second is the dispersion of injected ICG which might not be able to localize IPNs correctly. 

 Authors believe that PA technology combined with solidifiable ICG agar injection as a fiducial marker would overcome these limitations. As far as authors know, there are few studies on the utility of PA imaging to localize IPNs even if PA imaging is known to be useful for detecting sentinel lymph node or various small sized cancer such as breast cancer, melanoma or head and neck cancer. 

Authors tried to comprehensibly demonstrate the possibility of PA imaging technique for IPNs from in vitro, phantom to in vivo study. However, authors agreed that this study itself would lack to draw any definitive conclusion and further study using large animal such as rabbit or pig should be necessary to incorporate this technology into clinical settings. 

Authors made a schematic figure to help reviewers or readers understand how PA imaging with injected ICG agar as a fiducial marker would might be able to localize deeply located IPNs.

Figure 1. Possible clinical scenario to detect IPNs using photoacoustic imaging 

The following points need to be addressed: 

1) As the PA imaging still limited with depth (and the US imaging of lung is also difficult) how you think the study can be translated? 

Thanks for your comment. Maximum penetration depth on PA imaging could be affected by the frequency of US (ultrasound) transducer. In this study, 20-MHz linear array transducer was used that has optimum target depth of 10-15mm. If a low frequency US transducer would be utilized, a 3-5cm maximum penetration could be achieved like other study [17]. As you mentioned, one of concerns on PA imaging of lung would be US imaging of lung (especially inflated lung). But some researchers have developed thoracoscopic ultrasonography and have tested its efficacy with localizing sub-centimeter nodules in the porcine deflated lung as well as with obtaining sufficient sampling from lung tumors in the rabbit model (Wada H et al. Eur J Cardiothorac Surg 2016;49(2):690-7). So, authors believe that PA imaging incorporated into thoracoscopic US device in the near future` would be helpful to detect small sized nodules and ICG agar from deflated lung during operation. 

2) ‘Monomeric’ ICG vs J-aggregated ICG in agar: Preparation of J-aggregated ICG was already reported (Chemical Physics 1997,220,385-392). Could you please perform the characterization the latter species in agar? 

3) Then, as you suggest draining characteristics of ‘mono ICG’ was faster than ‘J-agar ICG’, could it be proved by incubating both current formulation and the J-agar pellet (a solidified agar+ICG) by incubating at 37C for 48 h to 76h. 

-> Thanks for your comment, advice and ideas. I have read how to make or prepare ICG J-aggregate (IJA) from some articles. But I could not perform and complete the characterization of IJA with agar due to time limitation and lack of facility in my lab. In next experiments using rabbit, I think that we will be able to complete the characterization of IJA agar and report them.

4) Overall the image quality is very poor; need to be post-processed to improve the quality; as well provide the spectra and graphs with clearly readable numbers. 

-> Thanks for your comment. The image quality has been improved and numbers in the spectrum and graphs has been changed to be readable after consulting with graphic designer. 

5) Is there any study on blank agar effect on surrounding tissue of injection, please cite the pertinent references. 

-> Thanks for your question. As far as authors know, there are two reports [19,20] on localization of small sized lung nodules using agar in human. Because there is no report on long term effect of agar on surrounding lung tissue because the tumor or injected agar should be completely resected during operation. However, Tuschida M et al. [19] reported that agar marker remained in place for more than 2 weeks after injection in animal models and there were no serious complications during and after injection of agar without any histologic deterioration.

 6) the phantom image, Fig. 2, extremely overlapped (high noise) does it cause signal spill-over, and over estimation. Can you comment? 

 -> Thanks for your comment. The PA signal spill over from high dose ICG (5mg/ml) was observed at shallow depth (less than 5mm) water media. But signal spill over was not observed when small tube was put into chicken breast muscle. As you can see Fig 2.B. the spectrum of PA signal from each concentration was not overlapped and smooth, which means that there was not spill over or overestimation. 

7) Where is statistical evaluation of in vivo analysis? 

-> In vivo study in this article was proof of concept. And due to the request of animal usage protocol, 3 mice were used in each time point which was too small to draw any statistical conclusions. 

8) Fig. 5. The depth scale is missing

-> The depth scale was added in Fig. 5. 

9) It is known that ICG not so photostable; comment whether other reported ‘stable’ PA chromophores could be suited for this study. 

 -> Thanks for your comment. As you mentioned, ICG itself is not so photostable, which means that ICG that was exposed to light could be photobleached with time. Authors tested ICG liposome, which is known to be more photostable than ICG, as PA chromophores. However, PA signal disappeared at the 10mm depth of chicken breast muscle probably due to the small amount of ICG (about 100ug/ml) in liposomal ICG. That is why author have used relatively high concentration ICG. Even if the photostability of ICG agar in phantom model was not performed in this study, strong PA signal from ICG combined agar was still detected 48 hrs after injection into mouse, indicating significant photobleaching did not occur. 

10) Is it possible to show, in an animal model, guided injection of agar+dye in pulmonary nodule. 

-> I added a photo showing US guided transthoracic injection of ICG agar into lung without any tumor. Even if lung tumor model in mice could be developed, small nodule could not be detected with US. That’s why ICG agar was injected into healthy lung without tumor. 

Reviewer #2: The main points of criticism of this manuscript pertain to (1) the confusion about the purpose of the study and (2) the unconvincing model systems used and the poor presentation of the data.

Ad 1) Even after reading the manuscript several times, it did not become clear to me which future application that preclinical study is supposed to prepare.

Is the ICG-containing agar supposed to represent a fiducial marker that is meant to aid surgical removal of suspicious nodules or is the idea that tumor cells could be labeled with ICG to facilitate their removal? For the latter case, the reader can only speculate on how selective targeting is supposed to work.

Thanks for your comment and question. To help reviewers and readers understand how PA imaging and ICG agar as a liquid fiducial marker for IPNs during operation, possible clinical scenario was added as follows. ICG itself has been being used for localization of IPNs (small lung nodules) during minimally invasive surgery. But ICG has a tendency to being dispersed and being washed out quickly, which would some drawbacks of ICG as a fiducial marker. To overcomes these limitation, ICG agar was made and tested using mice model. Because the fluorescence signal of ICG agar from more than 5mm depth would not be able to being detected due to scattering, the possibility of PA imaging for detection of ICG agar was explored in this study. 

Ad 2) Numerous photoacoustic imaging studies are using ICG-labeled cells in tissue phantoms and animal models; some of them come close to a setting that could eventually be relevant for clinical applications.

None of the data presented in the current work, however, comes close to a realistic scenario of detecting ICG-labeled cells or fiducial markers (see the confusion due to conflicting information in the text described in point (1)) within lung tissue, which is an intrinsically complicated tissue to image by photoacoustics due to the high density of air-filled structures.

While in NCI-H460 cells, ICG seems to be only detected by fluorescence, neither the data from blood vessel mimicking tubes nor the chicken breast is convincing, new, or convincing.

The in vivo work was then apparently just conducted with ICG-containing agar as opposed to cells injected subcutaneously or into the lungs of mice: the data quality is bad, and it is not clear what is supposed to be learned from the data other than confirming yet one more time that highly-concentrated ICG can be localized in tissue by photoacoustics.

Thanks for your comment. As you mentioned, several ICG labeled cells or nanoparticles were studied and reported. Authors also tested ICG liposome, which is known to be more photostable than ICG, as PA chromophores. However, PA signal disappeared at the 10mm depth of chicken breast muscle probably due to the small amount of ICG (about 100ug/ml) in liposomal ICG. That is why author have used relatively high concentration ICG. Even if the photostability of ICG agar in phantom model was not performed, strong PA signal from ICG combined agar was still detected 48 hrs after injection into mouse, indicating significant photobleaching did not occur.

As you mentioned, one of concerns on PA imaging of lung would be US imaging of lung (especially inflated lung). But some researchers have developed thoracoscopic ultrasonography and have tested its efficacy with localizing sub-centimeter nodules in the porcine deflated lung as well as with obtaining sufficient sampling from lung tumors in the rabbit model (Wada H et al. Eur J Cardiothorac Surg 2016;49(2):690-7). So, authors believe that PA imaging incorporated into thoracoscopic US device in the near future` would be helpful to detect injected ICG agar as a fiducial marker from deflated lung during operation.

In general, the presentation of the spectra and images look like distorted screen-shots from a poor GUI. The scientific images are missing defined colormaps and scale bars. Just as one example, how was the color-overlay in Figure 6B computed?

There are also many linguistic weaknesses in the text.

Thanks for your comment. The image quality has been improved and numbers in the spectrum and graphs has been changed to be readable after consulting with graphic designer. PA images was automatically overlaid to US image in the machine itself.

---

## [Decision Letter · Decision Letter 1]

11 Mar 2020

PONE-D-19-30111R1

Photoacoustic Imaging to Localize Indeterminate Pulmonary Nodules: A Preclinical Study

PLOS ONE

Dear Dr Lee,

Thank you for submitting your manuscript to PLOS ONE. After careful consideration, we feel that it has merit but does not fully meet PLOS ONE’s publication criteria as it currently stands. Therefore, we invite you to submit a revised version of the manuscript that addresses the points raised during the review process.

We would appreciate receiving your revised manuscript by Apr 25 2020 11:59PM. To enhance the reproducibility of your results, we recommend that if applicable you deposit your laboratory protocols in protocols.io, where a protocol can be assigned its own identifier (DOI) such that it can be cited independently in the future. For instructions see: http://journals.plos.org/plosone/s/submission-guidelines#loc-laboratory-protocols

We look forward to receiving your revised manuscript.

Kind regards,

Gabriele Multhoff, Prof. Dr.

Academic Editor

PLOS ONE

Additional Editor Comments (if provided):

Dear authors

I could not find a revised version of the Figures in a better quality. Please provide this in the next revision.

Reviewers' comments:

Reviewer's Responses to Questions

**Comments to the Author**

1. If the authors have adequately addressed your comments raised in a previous round of review and you feel that this manuscript is now acceptable for publication, you may indicate that here to bypass the “Comments to the Author” section, enter your conflict of interest statement in the “Confidential to Editor” section, and submit your "Accept" recommendation.

Reviewer #1: All comments have been addressed

2. Is the manuscript technically sound, and do the data support the conclusions?

Reviewer #1: Partly

3. Has the statistical analysis been performed appropriately and rigorously? 

Reviewer #1: No

4. Have the authors made all data underlying the findings in their manuscript fully available?

Reviewer #1: No

5. Is the manuscript presented in an intelligible fashion and written in standard English?

Reviewer #1: Yes

6. Review Comments to the Author

Reviewer #1: The authors attempted to address, however not satisfactorily, some of the questions raised earlier.

a) The quality of PA-images and spectra is not suitable for publication. If the visible quality of those images itself is difficult to perceive, how one can gain added value by using PA-image guided surgery in such ICG-agarose.

b) Manuscript lacks to bring out the advantage of using PA-image guided surgery.

7. PLOS authors have the option to publish the peer review history of their article (what does this mean?). If published, this will include your full peer review and any attached files.

Reviewer #1: No

---

## [Author Response · Author response to Decision Letter 1]

17 Mar 2020

Reviewer #1: The author attempted to address, however, not satisfactorily, some of the questions raised earlier. 

a) The quality of PA-images and spectra is not suitable for publication. If the visible quality of those images itself is difficult to perceive, how one can gain added value by using PA-image guided surgery in such ICG-agarose. 

Thanks for your comment. The PA image and spectra in current study were semi-automatically obtained from Vevo®LAZR system (Visualsonics, Fujifilm) and were submitted in this manuscript without any modification or manipulation to maintain the truth and reality of data. Unfortunately, authors was not able to get the PA images or spectra with more quality than those already submitted. 

b) Manuscript lacks to bring out the advantage of using PA-image guided surgery. 

I partly agree with your comment. As far as I know, this study is a first attempt to explore the possibility of PA imaging technology combined with ICG-agar to detect or localize small lung nodule. Because of size limitation, I was not able to demonstrate clinically relevant lung cancer model using mice to prove the advantage of PA image guided surgery in this study. Recently, authors are developing lung cancer model using rabbit for further study. We hope that these results will be reporting soon.

---

## [Editor Report · Decision Letter 2]

25 Mar 2020

Photoacoustic Imaging to Localize Indeterminate Pulmonary Nodules: A Preclinical Study

PONE-D-19-30111R2

Dear Dr. Lee,

We are pleased to inform you that your manuscript has been judged scientifically suitable for publication and will be formally accepted for publication once it complies with all outstanding technical requirements.

With kind regards,

Gabriele Multhoff, Prof. Dr.

Academic Editor

PLOS ONE
---

## [Editor Report · Acceptance letter]

9 Apr 2020

PONE-D-19-30111R2 

Photoacoustic Imaging to Localize Indeterminate Pulmonary Nodules: A Preclinical Study 

Dear Dr. Lee:

I am pleased to inform you that your manuscript has been deemed suitable for publication in PLOS ONE. Congratulations! Your manuscript is now with our production department. 

With kind regards,

on behalf of

Prof. Gabriele Multhoff 

Academic Editor

PLOS ONE